# The Promotion of Alternative Crop Production Paradigms Should Be Founded on Proven Science-Based Approaches

**DOI:** 10.3390/plants14050681

**Published:** 2025-02-23

**Authors:** Jillian Lenné, David Wood

**Affiliations:** North Oldmoss Croft, Fyvie, Tirriff AB53 8NA, UK; agrobiodiversity@gmail.com

**Keywords:** agroecology, VACS, sustainable food production, Global South, functional diversity, input reduction paradigm

## Abstract

Recent discourse on the need to adopt alternative approaches to sustainable crop production has strongly criticized modern, usually referred to as “industrial”, agriculture as the main cause of environmental problems and a loss of biodiversity, which is concerning given that modern agriculture currently feeds over 90% of the global population. Ongoing criticisms of modern agriculture have escalated into calls to replace it, largely based on the belief that alternative approaches will lead to sustainable crop production, although food production potential is rarely mentioned. This paper critically analyzes two such alternatives, agroecology and the Vision for Adapted Crops and Soils (VACS), a sub-set of agroecological approaches with a focus on the Global South. In the case of agroecology, this paper considers the role of diversity in cropping systems and the input reduction paradigm, as well as labor productivity. Modern agriculture already provides a wide range of science-based, proven crop and field diversification options without the need to rely on in-field crop species diversity. Furthermore, a reduction in or the elimination of chemical fertilizers with a preference for compost and manure is not a viable strategy where soils are severely degraded. In the case of the VACS, the planned emphasis on “opportunistic”, locally adapted, traditional/indigenous crops is challenged by the importance of introduced crops to food production, especially in Africa. We conclude by recommending a pragmatic approach to using all of the available agricultural practices wisely to produce enough food in an environmentally responsible manner. Global leadership is needed to bring the divergent views of scientists and environmentalists together to improve food production and nutrition, livelihoods, and the agricultural environment.

## 1. Context

In spite of the success of the Green Revolution (GR) of the 1960–1980s in substantially increasing the production of the staple food crops wheat and rice and improving food security and averting famines in the Global South [1] and the fact that the global crop production has increased by 390% while land cultivation has increased by only 10–15% in the past 60 years [2] due to science, technology, innovation, and entrepreneurship [3], the technologies on which these achievements were based—high-yielding varieties being grown in monocultures and increased inputs of fertilizers and pesticides—often referred to as “industrial” agriculture—have been strongly criticized [4]. The fact that continued post-GR yield increases based on genetic and agronomic improvements in these crops and their cropping systems globally, as well as in other important crops, such as grain legumes, root and tuber crops, and oil seeds, with measurable impacts on global food production [5,6,7], have reduced the overuse of fertilizers and pesticides seems to be ignored. Therefore, in spite of these achievements, criticism of modern technologies—often based on the over-exaggeration of negative environmental effects—continues, especially that by FAO [8,9,10] and transnational NGOs and activists [11,12]. It is true that some modern agricultural practices, especially in the Global North, including the continued overuse of agrochemicals and mono-cropping, are still causing environmental problems such as pollution, soil erosion, and a loss of wild biodiversity, and further efforts are needed to address them through proven, science-based approaches. Substantial research efforts are continuing to do so [13,14,15]. However, paradoxically, the main focus of the calls for the promotion of alternative crop production paradigms is for smallholder farmers in the Global South, where proven, science-based technologies still have much to contribute to increased food production [16,17].

Recently, these criticisms of modern agricultural technologies have escalated into growing calls for alternatives to replace “industrial” approaches based on a belief that alternative approaches will lead to an agricultural transformation and sustainable crop production. Interestingly, the potential for alternatives to produce enough food to adequately feed 8 billion people is rarely mentioned. Such calls seem to ignore the current fact that much of the global farming responsible for feeding over 90% of 8 billion people [3] is based on good agricultural practices, often referred to as GAP [18], such as improved, functionally diverse crop varieties usually in monocultures; crop rotation; mixed crop-livestock systems; the judicious use of agrochemicals; and integrated pest management, among others, combined with environmentally responsible approaches. For example, globally, mixed crop–livestock systems produce around 50% of the world’s food [19,20]. If such successful systems are to be replaced by alternatives, including agroecological approaches and traditional and/or indigenous crops [4,9,19,20,21,22], then they should also have a proven ability to produce enough food to feed the current population, as well as predicted future increases in the global population.

One could speculate that the successful promotion of such alternative methods in the Global South will perpetuate food insecurity and prolong the dependence on imported food from the Global North through “industrial” agriculture [23]. The stark reality is that an estimated 868 million people were moderately or severely food-insecure in Africa in 2022 [24]. As a result, Africa currently relies on imported food: food imports for African countries are 150% higher than their exports ([25], p. 51). Regrettably, activist organizations such as the Rosa Luxemburg Foundation ([26]; also see [23]) are communicating false information in regions where food systems urgently require improvement using existing modern technologies. The Global South deserves far better than these activists’ beliefs: it urgently requires food systems that have a proven ability to meet food needs and not alternatives with limited or no track records.

## 2. The Objectives of This Paper

The main objectives of this paper are to analyze two alternative trends to modern agriculture being promoted for future food production: agroecology and the Vision for Adapted Crops and Soils (VACS). We clarify that the main focus of the analysis is the promotion of selected agroecological approaches in the Global South. It is not regions where good agricultural practices are being promoted as agroecology, such as in Europe [27]. Throughout, we ask the following question—are alternatives needed to replace existing global crop production systems, or is the improvement of modern crop production systems a better option? In the case of agroecology, we briefly look at its history, especially its evolution from a science to a political movement, and then critically analyze several of its principles/elements: cropping system diversity and the input reduction paradigm being promoted to replace modern technologies and the importance of labor productivity. We focus largely on annual food crops, as more information is available on them. Further analysis is needed in perennial cropping systems. Similarly, for the VACS, we look at the recent promotion of traditional/indigenous and under-utilized crops as alternatives to the current staple crops based on their perceived local adaptation, nutritional qualities, and apparent greater resilience to climate change in comparison to the long-standing contribution of introduced crops to food production in Africa. We conclude that instead of a binary approach—replacing a global system with a proven potential to feed billions with unproven alternatives—a more pragmatic approach is needed to support the good agricultural practices which are most beneficial in terms of food production under particular circumstances—with the proviso that they should be science-based rather than based on beliefs. We also conclude that greater advocacy for the appropriate technologies is needed by scientists and policy-makers to counter the ill-informed and potentially dangerous calls for some alternatives to ensure that the Global South benefits from proven approaches to future food production.

## 3. Agroecology

### 3.1. The Historical Context of Agroecology

During the past 20 years, agroecology has grown in prominence within global agricultural discourse based on a belief that it can dramatically transform agrifood systems [4,28]. The concern of agroecologists is that modern, “industrial” agriculture has been principally responsible for agroecosystem degradation and should be replaced by agroecology. As there is no internationally recognized definition of agroecology [16], it is worth looking briefly at its history to try to understand what is currently being promoted as an alternative to modern agriculture. Wezel and Soldat [29] provide a useful account of the evolution of the term. From the 1930s to the 1960s, agroecology was firmly anchored in the science of crop production and protection. However, from the 1960s onwards, environmental activists, including Greenpeace [30] and La Via Campesina [31], among others, motivated agroecological campaigns, and agroecology evolved into a broad mixture of science, practices, and movements, with different groups placing varying emphases on these three main components, from the rational to the extreme. Definitions can change over time, but the significant broadening of the term agroecology and the explosion of different definitions are leading to growing confusion around what agroecology actually is [16]. On the one hand, agroecology allows many good farming practices to be labeled as agroecological, such as intercropping, crop rotation, and recycling animal manure, but, at the same time, rejects others deemed unagroecological without a clear scientific justification, such as monoculture agriculture, conventionally bred improved varieties, and GM crops and agrochemicals. A good description of agroecology is a “curate’s egg”—good in parts [32].

### 3.2. The IAASTD Report “Agriculture at a Crossroads”

In 2009, agroecology was internationalized in the International Assessment of Agricultural Knowledge, Science and Technology for Development (IAASTD)’s “*Agriculture at a Crossroads*” report [4], described as “*the biggest review ever of global agriculture*”. At a cost of USD 12 million, it called for priority to be given to traditional agriculture in the Global South based on agroecological approaches. It was sponsored by the World Bank, the Global Environment Facility, and the United Nations Food and Agricultural Organization (FAO), among others. This report spun off further reports and publications which emphasized and reinforced the need for a radical agricultural transformation in the following decade.

McIntyre et al. [4] made recommendations on profoundly transforming future food production as an alternative to modern agriculture. In essence, the replacement of modern agriculture, considered to be unnatural, vulnerable to pests and diseases, and unstable, would be based on ecological principles—more species diversity, the greater use of traditional (including indigenous) crops, reliance on vegetational diversity for pest and disease control, and no synthetic inputs (fertilizers or pesticides) or genetically modified (GM) crops, usually considered organic production—and, it was claimed, would therefore be more natural. This could also be described as “archaism”, the former state of agriculture prior to the application of formal agricultural research. It must be noted that the only model of modern agriculture highlighted for comparison was the “industrial” model—large, mechanized farms, high inputs of fertilizers and pesticides, and mono-cropping—with limited recognition of the many other models of modern agriculture practiced by large, medium, and small farms globally. It placed considerable unfounded potential in smallholder farmers meeting future food production needs, including highlighting complex, multi-functional, knowledge- and labor-intensive agriculture. An extensive critical commentary of McIntyre et al.’s [4] report has already been made by Wood and Lenné [33].

A year or so after the publication of McIntyre et al. [4], a critical evaluation of the IAASTD process and report was made by the Independent Evaluation Group (IEG) of the World Bank ([34]; Box 1). Serious concerns were raised about its credibility, legitimacy, cost-effectiveness, and potential impact both internationally and nationally [34].

Three countries, the USA, Canada, and Australia, endorsed the IAASTD report with notable reservations: “we have specific and substantive concerns”; “there remain numerous areas of concern in terms of balanced presentation, policies and… ambiguities”; and “we cannot agree with all assertions and options”, respectively (4—Annex G). As noted in Box 1, such concerns translated into a limited attributable impact at both the international and national level.

Box 1Key issues raised in the evaluation of the IAASTD process and report by the IEG, the World Bank [34]. The IAASTD was a useful experience at the nexus of politics and science. However, agricultural technology, with its complexity, diversity and politics, proved to be a bridge too far. It was a missed opportunity at considerable cost. Based on lack of adherence to the principles of openness, accountability and fairness, its legitimacy was also questioned. The overall message which emerged from the IAASTD was restrictive and exclusionary with an undercurrent against new technology and input-intensive agriculture. It highlighted a: “*radical transformation of the present chemical-based industrial and conventional food and agricultural production systems towards agroecology”* [34]. Although the objective of the IAASTD aimed not to be policy prescriptive, its advocacy of an extreme agroecological approach—a predominantly low-input, organic, smallholder farming, environmentally focused approach—was, in fact, prescriptive. And, more importantly with respect to sustainability, widespread adoption of the dominant IAASTD options would lead to reduced productivity gains and more environmental damage through extensification. The evaluation concluded that: “*So far, attributable impact at the international level has been modest at best, and at the national level and below, negligible”* [34].

Most importantly for both global and national agricultural policy, there was a serious problem of bias in the IAASTD report [4] with regard to future regional and global approaches to agricultural research and development. A search for the occurrence of the words “agroecology” and “agroecological” in the text of the five IAASTD sub-global reports produced concerning results (Table 1). The term ‘agroecological zone’ was not included, as it has a very different meaning to the current usage of ‘agroecological’ approaches. “Agroecological zones” are geographical areas with similar climatic conditions. This prior use of the word agroecological is now eclipsed by the current promotion of agroecology.

It is clear that the IAASTD global report [4] was dominated by a focus on the LAC sub-global report [35] in support of agroecology. Table 1 highlights 82.5% of the support for agroecology in the IAASTD sub-global reports was from just one region, Latin America and the Caribbean, out of five regions, some with far greater human populations and food needs. This clear disparity between regions is especially striking given the very low priority given to agroecology in the sub-Saharan Africa report [36] (a current major target for the promotion of *the Vision for Adapted Crops and Soils* project by the USA State Department, as analyzed below). Furthermore, 67% (36 of 54) of African countries did not endorse the IAASTD report [4], possibly because they did not want to erect barriers to accessing new technologies, such as climate-adapted and pest- and disease-resistant improved crop varieties, including GM crops. In fact, only 58 of 193 countries fully endorsed the IAASTD global report [4].

### 3.3. The On-Going Promotion of Agroecology

Since the IAASTD report [4] was published and in spite of the lack of interest in its messages, the promotional campaign for alternative crop production paradigms through agroecological approaches has continued to reemphasize and reinforce the need to move away from modern agricultural technologies, essentially promoting archaism [25,37]. This has been encapsulated in a set of principles [21] and/or elements [8].

The most relevant principles and/or elements in the context of sustainable crop production, namely diversity/biodiversity and input reduction/elimination, are summarized in Table 2. We then discuss the concerns raised about the implications of some of these for future food production in the Global South and the importance of labor productivity. It can also be suggested that the inclusion of other principles and elements which focus on food system equity and sovereignty ostensibly helps to legitimize these questionable technical principles/elements.

### 3.4. Diversity/Biodiversity (Element 1 and Principle 5)

The fundamental criticism of the central tenet of agroecology, the belief that diverse species mixtures are the best option for controlling insects’ herbivory of crop fields, ignores the Nature-based approach of the first farmers’ fields more than eleven thousand years ago [38]. Early agricultural fields in Western Asia were close models of Nature, in which monodominant vegetation of wild crop relatives was replaced with the earliest domesticated monodominant cereals—wheat and barley [38]. In excluding monocultures, agroecology has no concept of natural monodominance, although it exists fairly widely, particularly in regions where ecological stress removes non-adapted species [39,40]. Two key adaptations of the wild relatives of the earliest cereal crops in Southwest Asia were large seeds and awns, which conferred the ability for these large seeds to stay buried amid presumed dry-season vegetation fires [38]. The first fields were copies of monodominant Nature. This is contrary to the central tenet of agroecology: using species diversity to hide away from pest problems. In fact, the first cereals were able to grow as monocultures with sufficient resistance to prevailing pests and diseases.

Barrios et al. [9] designate biodiversity as one of the promising entry points to agroecology, while Herren et al. [25] consider diversity *“the mantra of agroecology”.* The priority given to diversity/biodiversity appears to be based on the belief that more diversity is ecologically based, will allow for the replacement of pesticides for managing pests [41], and will produce sufficient food and improve the livelihoods of smallholder farmers [26,42], although data supporting the latter are limited and/or are based on farmers’ perceptions [43,44]. Cropping system diversity should therefore replace monoculture agriculture, which produces most of our food but which is considered to be unnatural, ecologically dysfunctional, and vulnerable to pests and diseases [37].

McCouch and Rieseberg [45] explored the benefits, obstacles, and challenges in incorporating greater diversity into agroecosystems for improved sustainability. They posed a number of pertinent questions: “What kinds of diversity matter?”, “How do we manage crop diversity?”, and “How do components of diversity interact within or across systems and scales?” Such questions illustrate the complexities of incorporating diversity into agroecosystems and the need for considerable research effort to develop science-based solutions [46]. Furthermore, concerns have already been raised about the ability of diverse agroecosystems to produce sufficient food, manage pests, and improve livelihoods for smallholder farmers [17]. Additionally, the management of diverse agricultural systems is considerably labor-intensive, making them less attractive to smallholder farmers [47,48].

#### 3.4.1. Crop Diversity in Agroecosystems

Wood and Lenné [38] and Lenné and Wood [40] have demonstrated that cereal monocultures are an ancient method of farming founded in the origins of agriculture and that modern plant breeding generates and supports monoculture crops that have a broad genetic base and which are designed to be resistant/tolerant to prevailing biotic and abiotic constraints. In fact, although modern monocultures of improved crop varieties are morphologically uniform, they are demonstrably functionally diverse in their prevailing pest and disease resistance, nutritional traits, climate resilience (heat tolerance, drought resistance, etc.), and nutrient use efficiency. The inclusion of GM crops in such cropping systems adds another dimension to their functional diversity, with unique traits of disease and pest resistance, herbicide tolerance, and nutritional enhancement, among others, in a range of important crops (Box 2) [46,49,50].

Box 2Examples of functional diversity in modern crop varieties grown in monocultures. Over 40 years, the number of landraces of wheat used in crop improvement in CIMMYT (the International Maize and Wheat Improvement Center) increased six-fold, while over 30 years, the number of landraces of rice used in improved varieties by the IRRI (International Rice Research Institute) increased ten-fold [51]. The diversity of the spring wheat parentage in Canada, the USA, and Mexico has significantly risen [52]. IR 64, the most popular rice variety in Southeast Asia due to its high yield, early maturity, excellent cooking quality, and resistance to major diseases and pests, is grown over 50 million has in Southeast Asia alone. It is substantially diverse, with 79 parents, including 20 landraces, 1 accession of *Oryza nivara*, and 58 older varieties in its pedigree [53]. GM technologies have added further diversity through unique traits in disease and pest resistance, herbicide tolerance, drought tolerance, nitrogen use efficiency, and nutritional enhancement in a range of important African food crops, including maize, rice, sorghum, banana, soybean, sweet potato, and others ([50]: Table 1). Varietal mixtures provide a means to increase crop genetic diversity without the need for breeding and have been shown to improve stability and disease management [46]. Dual-purpose crops add diversity through combining quality and yield traits in food grain with the higher digestibility and protein contents of the fodder in crop–livestock systems [54]. The utilization of fodder from dual-purpose crops can minimize the use of external inputs, contributing to resource efficiency and a reduced environmental impact.

A number of studies in different cropping systems on different continents have concluded that the management of pests solely through diversity is inconsistent [17]. In a meta-analysis of crop pests and natural enemy response to landscape complexity, Chaplin-Kramer et al. [55] noted that consistent suppression of pests was not detected in complex landscapes over the time scales of their studies. Wyckhuys et al. [56] looked at the status and potential of conservation biological control (CBC) for 53 crops from 390 literature records, although most were for rice, maize, and cotton. They concluded that much more research was needed in most of these cropping systems. Furthermore, using a large database of 132 studies at 6759 sites worldwide, Karp et al. [57] found that natural enemy responses to pest abundance, predation rates, and crop damage were heterogeneous and inconsistent, and there was no consistent improvement in pest management. And recently, Wyckhuys et al. [58] studied the role of diversification with 25 legume species, commonly used as intercrops, for natural biological control in agroecosystems. Although the natural enemies regularly foraged on legumes, the data on the interaction linkages with pests were profoundly incomplete due to a lack of research on their mechanistic basis. Its scientific underpinnings were weak. Tropical, irrigated rice is the only cropping system where sufficient research has been carried out over four decades on understanding the role of diversity in pest management [17,59].

#### 3.4.2. Field and Farm Diversity

Crop biodiversity can be incorporated into fields in many different ways in time and space and at the field, farm, and landscape levels. These practices are not transformational, as claimed by agroecology [21,25], but rather have been part of farming systems, in some cases for centuries, since long before agroecology was conceived. Most have been the focus of decades of research by agricultural scientists and are common in modern agroecosystems globally. For example, intercropping, the planned combination of two or more crops in a field, is a widely used practice to increase biodiversity in agroecosystems and has been the subject of formal agricultural research since the 1890s [60]. As a comprehensive analysis was performed by Lenné [46], while key examples only are highlighted in Box 3. All of these can be considered good practices and part of modern agriculture, although they have been subsumed under agroecology.

Box 3Examples of field and farm diversity in modern agriculture. A meta-analysis of 226 field experiments showed that intercropping produced a diverse range of nutritional products with minimal yield sacrifice [61].Intercropping of grain legumes and cereals improves the use of soil N resources and reduces, but does not eliminate, the requirement for synthetic N fertilizer [62,63]. Strip cropping, a form of intercropping where two or more crops are grown adjacent to one another in long and narrow multi-row strips, can deliver a greater range of ecosystem services compared to monocultures due to increased interspecific crop interactions, spatio-temporal niche differentiation, and higher in-field habitat diversity [64]. Crop rotation of two or more crops with or without a fallow period has been used for thousands of years to increase the soil’s fertility and structure; reduce erosion; improve weed management; boost yields; contribute to pest and disease suppression; and increase the diversity of the cropping system and has been subject to many years of formal agricultural research long before agroecology [65,66,67]. Mixed crop–livestock systems produce about 50% of the world’s food [19,20] and are common in both developed and less developed countries. Such systems offer circularity of the nutrient flows [68], enabling farmers to integrate crop production, which provides food grain and crop residues as feed, with the livestock providing manure for fertilizer. Crop–livestock integration has been especially effective in promoting resilience in tropical America [69].

Moore et al. [70] explored a range of multi-cropping options, including intercrops, strip crops, relay crops, cover crops, etc., for increasing the diversity temporally and spatially in fields in the context of plant breeding. Varietal development for such multi-cropping fields requires a systems approach, which includes agronomic and crop ecology knowledge to inform the breeding decisions. The specific breeding objectives for multi-crop systems will depend on the cropping system, including the component crops, the needs of the farmers, and numerous environmental and management factors influencing the competitive and cooperative interactions among crops. Much more research is needed on these important issues to develop improved varieties for multi-cropping systems.

### 3.5. Input Reduction and Recycling (Principles 1 and 2 and Element 4)

The promotion of input reduction and recycling by FAO [8] and the HLPE [21] calls for a reduction in or the elimination of external inputs (inorganic fertilizers) and the greater use of local renewable resources through the recycling and composting of biomass and animal manure (Table 2). The majority of crop–livestock farmers in the Global South already recycle animal manure [68], although this does not ensure the resulting composted biomass will provide sufficient nutrients for adequate crop growth. Often, there is not enough compost, and the nitrogen and phosphorus levels in the manure do not match the nutrient needs of crops. Farmers must use chemical fertilizers to supplement the nutrients from manure for adequate food crop production.

In a comprehensive review, Falconnier et al. [63] stressed that the reliance of arable farmers on local composted renewable biomass and nitrogen fixation by legumes instead of chemical fertilizers is highly inappropriate for smallholders in the Global South, especially in Africa, where most soils are degraded. The reality is that the average per-hectare chemical fertilizer application in Africa is 16 kg—15% of that in Latin America and 10% of that in South Asia [47]. As expected, Falconnier et al.’s [63] analysis showed that *more* chemical fertilizer is needed in Africa after decades of crop production with very low levels of fertilizer leading to widespread soil nutrient mining. The nitrogen needs of crops cannot be adequately met solely through biological nitrogen fixation by legumes and the recycling of animal manure. Chemical fertilizers, if used appropriately, cause little harm to the environment. Most importantly, reducing the use of chemical fertilizers would hamper productivity gains and contribute indirectly to agricultural expansion and deforestation.

The main implication of adherence to Principles 1 and 2 and Element 4 (Table 2) is the adoption of the alternative production method of organic farming. But is this appropriate for smallholder farmers in the Global South and for ensuring food security? Meta-analyses summarizing a wide range of comparisons concluded that organic yields are overall 19–25% lower than the yields in conventional agriculture, although the yield gaps may vary depending on the crop and management practices [2,43,71,72]. A yield gap of 25% implies that 25% more land is needed to produce the same amount of food, which is problematic given the negative consequences of expanding agricultural land use [2]. This ratio is not, however, appropriate for on-farm practice for two reasons [73]. First, organic yields are smaller than those in experiments because organic growers do not have tools for the control of weeds, diseases, and pests that are as effective as those that researchers have. Second, and more importantly, in organic production, some land must be allocated to legumes to provide nitrogen for the entire system [73]. An effective organic/conventional production ratio on farms is closer to 0.5. Herein is the major disadvantage of the expansion of organic agriculture for food production in the Global South. There is no scientific evidence for the validity of the concept of excluding mineral fertilizer in crop nutrition [71]. The rejection of mineral fertilizers is often treated as a doctrine, and proof is seldom requested.

Two recent examples of the wide-scale conversion of conventional agriculture into organic agriculture—Cuba and Sri Lanka—highlight the realities of trying to achieve sustainable crop production through the elimination of chemical inputs (Box 4).

Box 4The impact of the replacement of agricultural chemicals with organic production in Cuba and Sri Lanka. In the 1990s, Cuba was forced to return to pre-industrialized farming methods due to the collapse of the Soviet Union and a loss of access to subsidized fuel and agricultural chemicals. Agroecology farming methods (oxen, hand hoes, animal manure, intercropping, and natural biocontrol of pests) promoted by NGOs such as La Via Campesina replaced previous conventional methods [47]. In 2014, twenty-five years after the enforced agroecology experiment, Cuba produced 37% less food than it had in 1990. Although Cuba is currently reviving its conventional farming sector with help from Brazil, it currently relies on food imports. In April 2021, the Sri Lankan government imposed a ban on the import and use of agricultural chemicals (fertilizers and pesticides), largely based on advice from Vandana Shiva, an influential Indian activist [74,75]. The ban contributed to a 20% reduction in rice production and the need to import USD 450 million of rice (Sri Lanka was previously self-sufficient in rice) and led to a surge in food prices. In November 2021, the Sri Lankan government partially reversed this policy, allowing the import of chemical inputs for critical export crops such as tea. Despite the reversal, the subsidies for chemical fertilizers were not reinstated, and food prices remained high and in short supply, as yields for major crops such as rice had not recovered.

The main reason for the promotion of organic agriculture is that it is believed to achieve better environmental impacts per unit of land used, although there is considerable uncertainty about its environmental performance [76]. And recent analyses have provided further clarity. Although nitrogen fertilizer field applications generate emissions of 1.13 Gt CO_2_e annually, manure and organic fertilizers add another 1.0 Gt CO_2_e emissions annually due to the emissions from composting biomass and manure [77]. Within the bounds of errors, there is very little difference between the two. Efficient nutrient use in both systems can help mitigate emissions through improved practices and technologies. More intensive agriculture might be the “least bad” option for feeding the world while saving its species—provided the use of “land-efficient” systems prevents the further conversion of wilderness into farmland [78], which would occur if organic farming were expanded.

### 3.6. Labour Productivity: A Forgotten Aspect of Agroecology

The adoption of both the diversity and input reduction elements/principles of agroecology will result in more labor-intensive practices for smallholder farmers. This is acknowledged by MacIntyre et al. [4], Smyth [23], and Herren et al. [25]. Belay [26] notes that *“The labour involved in agroecological farming practices often leads to misconceptions of agroecology as a backward step to the labour-intensive practices of the past. In reality agroecology involves labour of a different kind—a kind that is intellectually engaging and physically rewarding”.* We must wonder how many smallholder farmers contributed to this conclusion. Paarlberg [47] concluded that agroecological farming methods require far too much human labor to be attractive to farmers once they have gained access to modern agricultural technologies. He cites a study on waru waru farming (raised beds with hand-planting, hand-weeding, and hand-harvesting) in the Andes of Peru which found the production costs including labor were USD 480 for each 11.2 kg of potatoes. Of most importance, Cock et al. [79] highlighted that the gap in labor productivity between the Global North and the Global South is now much greater than the yield gap. This large labor productivity gap, unless remedied, will (i) condemn many farmers in the Global South to live in poverty and (ii) make them less competitive and force them to follow the well-established trend of exiting farming altogether, which (iii) will contribute to a greater dependence on imported food in many countries.

## 4. The Vision for Adapted Crops and Soils (VACS): A “Sub-Set” of Agroecological Approaches

The recent promotion of *the Vision for Adapted Crops and Soils*, known as the VACS [22,80], emphasizes revitalizing “opportunity” crops variously defined as indigenous/traditional /under-utilized crops in Africa for more nutritious and climate-resilient cropping systems. This vision is based on the assumption that these “old” crops are “***better equipped** to provide stable and nutritious diets in the face of climate variability and extreme weather events in geographies across the continent*”. The evidence for this assumption is weak. Why should African traditional/indigenous crops be more nutritious and better adapted to climate change than the diversity of introduced crops that have replaced them and that are the mainstay of food security in Africa and, in fact, globally? Both Wood [81] and Khoury et. al. [82] have shown that around 70% of the crops grown worldwide are introduced. Over 90% of the food crop production in Malawi is from introduced crops. Currently, introduced crops are the foundation of global food systems. The emphasis on **“better equipped”** seems to be based on the widespread belief that these crops are “locally adapted”, but it ignores the fact that traditional/indigenous crops are also, and inevitably, locally constrained by their long contact with co-evolving pests and diseases [83,84,85,86,87]. Past studies of the pathogens in populations of wild relatives of crops have shown the adaptation of pathogens to local hosts [88].

The VACS is a sub-set of the agroecological approaches described in the IAASTD report [4] and subsequently promoted by FAO [8] and the HLPE [21]. For example, it states, *“The challenge now is to acknowledge and promote the diversification of production systems through the domestication, cultivation, or integrated management of a much wider set of locally-important indigenous species*” ([4]: p. 223). This ignores the fact that small-scale farmers want the *best* crops in terms of nutrition and food security, rather than diverse mixtures of species. It also ignores the considerable levels of functional trait diversity for nutritional enhancement and climate adaptation already bred into higher-yielding, improved varieties of important staple food crops [40]. Crucially, it ignores the value of crop introduction, allowing crops species to escape co-evolved pests and diseases in their continent of domestication [81,85,86].

The VACS “vision” is one shared by the USA Department of State, whose motto is that it leads *“America’s foreign policy to advance the interests and security of the American people*” [89]. This vision is overtly top-down. In particular, the draft list of crops identified by Karl et al. ([22]: Table 5) does not appear to have been developed with African countries but by the Centre for Climate Systems Research (AgMIP) at Columbia University, New York, through modeling and AI based on climate issues but apparently not nutritional value. There is no evidence that farmers, consumers, or rural communities in Africa were consulted about the inclusion of these crops.

The basic error in the VACS’s belief in “local adaptation” (that local, indigenous crops are somehow better adapted to that particular environment) seeks to prevail over the importance of introduced crops to staple food production. Fundamentally, the value of the crop introduction strategy dates back to the Columbian Exchange after the discovery of the Americas, when hundreds of thousands of transatlantic movements of seed of thousands of varieties of hundreds of crops occurred (see 86 for a comprehensive review). By the 19th century, systematic crop introduction had been established in the Global North, while the Global South reaped its benefits in the 20th century. As was noted by Masefield [90], “*If there was one thing that had been clearly shown by the experience of the nineteenth century, it was the potential value of crop introductions from one country to another. By 1900 this had become almost an article of faith rather than of policy, and this activity was the main preoccupation of many of the new Departments of Agriculture...*”. Note that the “article of faith”, the escape from co-evolved pests and disease, was not initially known. Most importantly to future food production, a major feature of modern agriculture relies on the introduction of crops to remove species susceptible to co-evolved herbivorous insects and diseases through long-distance dispersal [83,84,85,86]. Box 5 illustrates selected examples of the unparalleled value of crop introduction.

Box 5Selected examples of the value of crop introduction. The “*most important general rule for enhancing crop productivity is introduction of economically valuable plants to suitable alien environments without the concomitant introduction of their native pests and diseases.*” [91]. “*The success attending the introduction of the soybean is without parallel in modern US agricultural history*” [92]. It is ironic that the USA’s crop production and associated export is almost 100% based on introduced crops (the exception is the local sunflower (Helianthus annuus) [83]. The uptake of introduced crops can be extremely rapid—for example, soybean in Brazil [93] and new varieties of wheat in India [94]. The benefits of escape from pests also extend beyond crops to pasture species, such as the fodder grass *Brachiaria brizantha*, introduced from Africa into Brazil, where it occupies, as a monodominant, around 50 million has [17].

The clear evidence that crop introduction allows for an escape from co-evolved pests and diseases is a major warning for the VACS. As the VACS has chosen “indigenous and traditional” opportunity crops as a target for future food production in Africa, these crops will be constrained by local pests and diseases. The expansion of indigenous crop production in Africa risks increased problems with indigenous pests and diseases, with a very limited knowledge base on how to deal with them. This is therefore a threat to future food production, resulting in African countries becoming increasingly reliant on food imports.

## 5. Concluding Remarks

One of the main messages emerging from the ongoing debate on an agricultural transformation for future sustainable crop production is that the current food production systems are broken and urgently need to be replaced, in spite of the reality that they produce most of our food. Transnational activists and UN agencies appear to be convinced that alternatives such as agroecological approaches and traditional/indigenous and/or under-utilized crops will more successfully achieve food security in the Global South than proven science-based approaches, including monocultures of improved, introduced crops and integrated soil fertility management. As we have shown in our comprehensive analysis, the inevitable result of such a transformation in countries in the Global South would be reduced food production, resulting in greater reliance on imported food crops.

In asking the question of whether alternatives are needed to replace existing global crop production systems or whether a better option is the improvement of modern crop production systems, we have emphasized the critical importance of national and regional staple food production. Trade-offs are likely to be necessary, or alternatives such as agroecology will require more land, with an associated biodiversity loss to meet the food demands of growing populations. Rather than the binary conflicts we are currently experiencing, a more pragmatic approach to using all available agricultural practices wisely to produce enough food in an environmentally responsible manner is needed. The best conventional and agroecological approaches are well suited to integration. Some examples include the intercropping of improved, pest -and disease-resistant, and climate-adapted varieties or monocultures of GM pest-resistant crops with conservation or regenerative agricultural practices, to name just two opportunities. We conclude that greater advocacy for the appropriate technologies is needed by scientists and policy-makers to counter the ill-informed and potentially dangerous calls for certain alternatives in future crop production to ensure that the Global South benefits from proven, science-based approaches to its future food production. Global leadership is needed to bring together the divergent views of scientists and environmentalists to improve food production and nutrition, livelihoods, and environmental sustainability.

## Figures and Tables

**Table 1 plants-14-00681-t001:** The number of times agroecology/agroecological is mentioned in the IAASTD global and sub-global reports.

Global and Sub-Global Reports *	Use of the Terms Agroecology and Agroecological
Global	71
LAC	174
ESAP	13
NAE	13
SSA	4
CWANA	7

* LAC = Latin America and the Caribbean; ESAP = East and Southern Asia and the Pacific; NAE = North America and Europe; SSA = Sub-Saharan Africa; CWANA = Central and West Asia and North Africa.

**Table 2 plants-14-00681-t002:** Selected agroecology principles/elements and their implications [8,21].

Principles/Elements	Implications
**Element 1** Diversity: “diversification is key to agroecological transitions to ensure food security and nutrition while conserving, protecting and enhancing natural resources” **Principle 5** Biodiversity: “maintain and enhance diversity of species, functional diversity and genetic resources and thereby maintain overall agroecosystem biodiversity in time and space at field, farm and landscape scales”	Focus on traditional and indigenous crops with unproven food production potential; The rejection of introduced crops and improved crop varieties with critically important traits, including GM crops; In-field diversity with the complexities of planting, managing, weeding, and harvesting mixed crop stands; Inconsistent pest and disease management through diversity; Nutritional diversity based on limited evidence; Ecosystem services and climate change resilience based on limited research; Re-classifying modern agricultural practices such as intercropping and crop rotation as “agroecological”.
**Principle 1**. Recycling: “preferentially use of local renewable resources and close as far as possible resource cycles of nutrients and biomass” **Element 4.** Recycling: “recycling means agricultural production with lower economic and environmental costs; crop–livestock systems promote recycling of organic materials by using manure for composting or directly as fertilizer” **Principle 2**. Input reduction: “reduce or eliminate dependency on purchased inputs and increase self-sufficiency”	Reductions in or the elimination of inorganic/chemical fertilizers; Reliance on organic fertilizers through recycling and composting of local renewable resources, including manure, in crop–livestock systems; Lower-yielding organic agriculture; Extensification.

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
