# Peer review of "The Promotion of Alternative Crop Production Paradigms Should Be Founded on Proven Science-Based Approaches"

_plants, 2025, doi:10.3390/plants14050681_

Round 1

Reviewer 1 Report

Comments and Suggestions for Authors

See attached file

Author Response

Responses to reviewer 1:

Review Lenné & Wood. I thoroughly enjoyed reading this paper and found the arguments and conclusions well presented and sound. The paper, is as would be expected, well written and pleasant to read. The paper is timely as, and here I totally agree with Lenné and Wood, the whole area of agroecology has become a fad that is promoted on the basis of emotions rather than facts. Similarly, the whole basis of VACS plays on the emotions that there is something wonderful about traditional crops grown in their centre of origin, which goes against all the accumulated knowledge and experience of agricultural scientists who listen to the facts, and the realities of global agricultural food production.

Thank you for your positive and well-received feedback on our paper. It is much appreciated.

I have some quibbles, at least to justify this review! Most of these are in a rather disjointed format as I made them as I went through the paper. I will make a short summary of what are my main concerns here which is followed by the more detailed comments below.

  1. I question the statement that globally mixed crop-livestock systems produce around 50% of the world’s food (18). This may well be one of those myths like two which Lenné recently debunked most effectively. This should probably be left out as it will not detract from the document, and one would hate to see doubt cast on the veracity of this paper as a result of this one dubious statement.

Actually this is not a myth. The paper was authored by livestock scientists and economists from 5 CGIAR centres as well as ACIAR and the World Bank. Authors include John Lynam, Carlos Sere, Mark Rosenberg and Stanley Wood – all noted economists. It is the foundation paper on this topic. Subsequent papers have supported these findings (see below) but we prefer to cite the foundation paper.

Subsequent papers:

Herrero M, Havlík P, Valin H, Notenbaert AM, Rufino M, Thornton P K, Blummel M, Weiss F, Obersteiner M (2013). Global livestock systems: biomass use, production, feed efficiencies and greenhouse gas emissions. PNAS 110 (52), 20888-20893.

 Herrero M, Thornton PK, Power B, Bogard J, Remans R, Fritz S, Gerber J, Nelson GC, See L, Waha K, Watson RA, West P, Samberg L, van de Steeg J, Stephenson E, van Wijk M, Havlik P (2017). Farming and the geography of nutrient production for human consumption. The Lancet Planetary Health 1, 33-42.

Kerr RB, Hasegawa T, Lasco R, Bhatt I, Deryng D, Farrell A, Gurney-Smith H, Ju H, Lluch-Cota S, Meza F, Nelson G, Neufeldt H, Thornton P (2022).  Chapter 5: Food, Fibre and Other Ecosystem Products. In: Climate Change 2022: Impacts, Adaptation and Vulnerability

https://www.ipcc.ch/report/ar6/wg2/chapter/chapter-5/

These clearly show the fact that most crop- and livestock-derived nutrients are produced in places where the systems are mixed – and in lower-income countries, that proportion is even higher.

Too often, foundation papers are ignored/not known about which leads to repetition of previous work – often claimed as novel.

  1. Throughout the paper I got the impression that the short season crops were emphasized, and perennial crops were largely neglected. This is a pity as in the Global South there are several perennial crops of great importance.

Yes, we did focus on major annual food crops as there is more information in the literature. There is limited information on agroecological approaches to perennial cropping systems e.g. sugarcane, bananas etc. with the exception of agroforestry systems seemingly dominated by World Forestry (ICRAF) where most of the perennials are not food crops. There is a need for a paper that focuses on perennial crops. However we will look for opportunities in the paper to refer to perennial crops.

  1. The main criteria for evaluation of the distinct technologies appears to be yield taken as yield per ha which has become synonymous with productivity. Farmers are often more interested in the profitability per ha and the labour productivity. This does not seem to be considered. For example, when discussing mixed cropping (within the field) difficulties of crop management and labour productivity are not mentioned, however from personal experience I know it is much more difficult to manage and work in mixed crops and labour productivity tends to be much lower.

Yes, we do agree. Agroecological cropping systems based on diversity and input reduction are much more labour intensive than conventional cropping systems and it is expected that labour productivity will be less. We have now included this in the paper with several references including: Cock et al., (2022) Labour productivity: the forgotten yield gap. Agricultural systems 201: 103452;  a Paarlberg paper and another by Daum et al. 2023 as well as a paper by a pro-agroecologist who considers the labour required for practising agroecology to be intellectually engaging and physically rewarding! Clearly he is not doing it himself.

“The labour involved in agroecological farming practices often leads to misconceptions of agroecology as a backward step to the labour-intensive practices of the past. This is a myopic view. In reality agroecology involves labour of a different kind — a kind that is intellectually engaging and physically rewarding”.  

  1. I find throughout the paper little mention of the producers or farmers´needs, this despite a statement that “Furthermore, concerns have already been raised about the ability of diverse agroecosystems to produce sufficient food, manage pests and improve livelihoods for smallholder farmers”. For example the document states: This ignores the fact that small-scale farmers want the best crops for nutrition and food security, rather than diverse species mixtures”. I simply do not believe that farmers want the best crops for nutrition and food security. Surely, they want crops that improve their livelihoods! This lack of attention to the real needs of farmers and the assumption that the role of farmers in this world is to provide food security and nutritious food, largely for others, certainly will not lead to ensuring food security. Somewhere this lack of attention to the legitimate needs and the role of farmers, who after all produce all the food, in food security should be addressed. To me one of the biggest failings of the Agroecology movement is precisely its lack of attention to the realities that farmers face. I personally believe that if we don´t give farmers what they want and need we´ll never have food security.

There is a problem with this argument. Agroecologists would argue that the best cropping systems for improving farmers’ livelihoods are diverse – and the more the better. Many smallholder farmers that we have spoken to in Latin America, Africa and Asia say they want crops/varieties better than the ones they are currently growing – whatever the cropping system. If this translates into improving their livelihoods then this is even better. We agree – we need to change the wording slightly but by arguing based on livelihoods we open up a case for agreeing with agroecologists obsession with diversity.

  1. One of the strongest arguments for questioning VACS is the escape from diseases and pests. The arguments are well presented, but in the conclusions. With the paper structured as at present the arguments should be brought forward and in the conclusion it should simply state that this concentration on traditional crops to be grown in their centre of origin is a fatal flaw in the program.

We have made sure that they key arguments for questioning VACS based on escape from pests and diseases are now presented in the VACS section.

  1. I think the lumping together traditional and underutilized is mistaken. Under utilized and grown outside the centre of origin where diseases and pest co-evolve are two distinct properties of the crops. They should be treated separately. There is surely a good case to be made for working with underutilized crops, but they may be most useful outside their centre of origin.

This is the main problem with the current version of VACS. It lumps traditional/under-utilized/ indigenous together and uses all three terms interchangeably as “opportunity” crops. Even though VACS has a Table of the potential crops to include – it has not yet been finalised. This Table includes all categories including 9 CGIAR mandate crops which are neither traditional nor under-utilised although several are indigenous in Africa – sorghum, millet and cowpea. We need to better explain this in the text.

Line 16. I would eliminate especially Africa. This is a problem anywhere with degraded soils and there are quite a few outside Africa. What is true is that in Africa in general fertilizer use is very low and nutrient response tends to be very high.

Changed

Line 24. I would not include maize here. The green revolution was initially basically wheat and rice. Maize, at least in the global south, began to move much later and here the authors refer to the 60-80s.

Changed

Line 26. with land use increasing by only 10-15% in the past 60 years. Is use the right word. Surely it is better to use cropped land or if you include pastures or range then agricultural and range land use or something like that. I suspect however that the authors are referring to cropped or cultivated land area.

We have checked the reference and clarified.

Line 27-28. I would mention that it is not just varieties, fertilizers and pesticides but also planting at higher densities, herbicides (OK maybe included in pesticides) and lots of irrigation schemes. Also, there was a procession from the first high yielding pest and disease susceptible varieties in rice towards host plant resistance, starting with IR20 and 22 if I remember rightly and later IR64 which is mentioned, which greatly reduced the use of pesticides. This to me is very important to bring in early when discussing industrial versus agroecology as it shows that proper deployment of improved materials in monoculture can be very effective.

Agreed – reduced use of pesticides and fertilizers needs to be included although this is alluded to in the section on in-crop functional diversity. Understanding of the in-built pest management characteristics of irrigated rice systems is critical to this. If you grow IR64 in non-irrigated systems – it is a different story.

Although not a green revolution crop, sugarcane in much of the world has extremely low pesticide usage due to both such practices as host plant resistance, bio control and generally good management practices such as clean seed. To me this is one of the best examples of monoculture with often no crop rotation with large areas of contiguous fields.

Good point – we will see where we can mention perennial crops.

Line 51. For example, globally mixed crop-livestock systems produce around 50% of the world’s food (18). This is one of those estimates that are happily bandied around with little attention to their validity. Lenné is well aware of this problem as she recently questioned the myth of 70% of world food from small holders. I would suggest great caution with this figure unless the authors can back it up with more than reference 18 which in turn refers to a report from ILRI (note ILRI is likely to be biased towards livestock as that is what they do or did!) which I was unable to access even though I tried quite hard to find it. All I could find was a Power point presentation with no details. Just a bald statement.

As we explained above, this is a valid reference not only authored by ILRI but 4 additional CGIAR centres including 4 noted economists and supported by several subsequent studies. We will cite one of the more recent studies. As noted above, the contribution to food production is even higher in less developed countries.

It is also not clear what is meant by mixed crop livestock. Are pigs or poultry reared on crop based diets part produced miles away part of a mixed crop livestock system? More clarity is needed over this point if the authors wish to stress it. My feeling would be to leave it out as in the preceding sentence the mixed crop livestock systems are mentioned. Note: later this figure is mentioned again.

It is an important reference to indicate to agroecologists that crop-livestock systems are a common cropping system which has been in existence for hundreds of years, make an important contribution to global food production, nutrition and food security and are diversified. Agroecologists have claimed crop-livestock systems as their own rather than being part of farming systems for hundreds of years.

Line 60. As a result, Africa currently relies on imported food: food imports for African countries 60 are 150% higher than exports (24, pg. 51). The authors might like to point out that Africa has moved from being a net exporter of food after World War II to an importer of food and feed.

It is true that food security has got worse in Africa in recent years. This is most likely more attributed to the impacts of climate change and political unrest rather than the prevailing agricultural systems. We are reluctant to open up this issue as it will give agroecologists further justification for criticising current science-based agricultural technologies – without proof but agroecologists do not seem to think that proof is needed. Also most of Africa’s major exports post WW2 were cash crops such as tea, coffee, cocoa, cotton, tobacco, rubber, sisal etc. The statement clearly needs greater detail but we are using it to indicate that Africa needs to produce more food in-country/continent or it will continue to rely on imports.

Line 63. The food insecure in Africa deserve far better than activist’s beliefs: they urgently require food systems that have proven ability to meet food security needs not alternatives with limited or no track records. I would suggest that the poor smallholders also deserve something better than the ravings of the sandal and bearded ecowarriors. The authors may decide not to mention sandals and beards, but I would suggest that they do mention the farmers who are an integral part of food security, although all too often that is forgotten. (This reviewer has a bee in his bonnet about the lack of attention to the livelihoods of those that work on the land and in food production when discussing food security). No line in particular. When this reviewer was a young student (a long time ago) he attended courses on Good Husbandry. So many of the supposedly newfangled ideas of agroecology are merely what I learnt as good husbandry as a farm boy and later at university! Hence, I totally agree with much of what the authors say about much of agroecology being old hat. Maybe the authors would like to mention the fact that good husbandry has long been part of and parcel of good farming practices.

We fully agree – but do need to be careful with emotive language as the other reviewer pointed out – and I thought we were being over-cautious already. Some of these issues have already been pointed out in other papers e.g. Lenné (2023) Current agricultural diversification strategies are already agroecological. Outlook on Agriculture 53, 271-280. 10.1177/00307270231199796 which is a comprehensive review of the fact that agroecology has taken many of the existing successful good agricultural practices and claim them as their own. We will emphasize this more in the paper.

Box 3. Ignominious 50% mixed livestock raises its head again.

See above comments – we wish to keep this reference as it is the foundation paper.

Around line 267. Inter-cropping within field is incredibly difficult to manage. Weed control is often a nightmare. Mechanical harvesting is very difficult. A further problem is that yield is normally the measure of success used by researchers when evaluating these systems. Labour productivity rather than yield is often more important to the farmer. Few researchers measure this but ask anyone who has managed intercropped fields and you´ll find that they know labour productivity is much lower. The authors should mention this problem within field intercropping and suggest that labour productivity should be taken into account as well as yield when evaluating systems. Note: strip cropping can obviate many of these problems.

We have now included labour productivity – not only for intercropping systems but for other diversity-related systems and recycling compost and manure being pushed by agroecologists.

Lines 299-313. I have always used as a sort of rule of thumb that about 2-2.5 t cereal equivalents per ha is about the maximum you can get out of a system without external plant nutrient supply on a long-term basis. I prefer this to the 2-3 ha to produce what 1 ha produces from the reference to Connor. The problem with the 2-3 ha approach is that the agroecologists will take this and say that in the US the average maize yield is about 11 tons per ha so with the agroecology systems you can get 5.5 tons on a long-term basis. Well, you can´t! I suspect that in the Loomis & Connor book on Agricultural ecology they mention this. I certainly remember talking about it with Loomis many years ago. I would suggest that if you can dig out an absolute figure for the cereal equivalents it would be better than the 2 3 ha.

I sought further feedback from David Connor who I know well. This refers to the need to have available 2-3 ha additional land to produce the biomass to produce composted nutrients equivalent to the recommended chemical fertilizer for 1 ha of crop production. This means that you need additional land which was the main reason for including this in the context that recycling on-farm biomass to replace chemical fertilizer is not a viable option in the long term for smallholder farmers. We do not want to get into a debate with agroecologists about using the USA as an example.

Line 341. It has long been known that introduced crops are important for several reasons. The great Purseglove (who you refer to) may have been the first to comment on their importance. Later Jennings and Cock (1975) [I know that in the guidelines one is not supposed to refer to oneself, but I really do believe that in this case this paper should be cited as it is highly relevant to the discussion and was the first that I know of to provide data that supported Pursegloves observations] provided data to show that yields were generally greater outside the centre of origin with one of the advantages of exotic crops being that they tend to have less diseases and pests that attack them (at least until someone illegally imports plants harbouring the diseases and pests). They then stated categorically “To the extent that food preferences and nutritional quality are not serious considerations, developing nations might better emphasize the production of introduced food crops. For the reasons presented, the Americas would be better advised to stimulate rice, sorghum, taro, cowpea, and soybean production. Africa and Asia would find greater success with maize, dry beans, cassava, potato, and groundnuts. This strategy for selected regional stimulation of basic food crops would duplicate the existing situation for most industrial crops including pineapple, banana, sugar- cane, sunflower, and the tree crops described by Purseglove”. The recent increases of cassava production in Asia and in Africa and beans in Africa are good examples of how this strategy has been successful with international R&D efforts in beans skewed to Africa and cassava to Asia and Africa. Oil palm is another wonderful example which I will return to later. I would suggest that some mention be made of this disease and pest avoidance of introduced crops, which surely fits with all the supposed philosophy of the agroecology crowd, and yet they fail to recognize the fact. Having read further on the disease and pest avoidance is mentioned, but surely it should be emphasized more possibly as a key point showing that emotion-based strategies dominate over science and fact-based strategies. As I later came to the conclusions I found more development of the theme of disease escape. It seems to me that this theme is of vital importance and all the evidence is that introduced crops at least have a window of opportunity have a window of opportunity for relatively disease and pest free cultivation when compared to their native conditions. Surely this is one of the clearest cut cases where one can say that the agroecologists are letting their emotions suppress the watertight evidence. All this should not be left to the conclusions, the arguments should be made earlier and then in the conclusions it should be clearly stated that the insistence on traditional (ie not local) materials does not make sense.

We agree – we did cite Jennings and Cock later in the paper but we need to develop this theme further earlier – thank you – this has been a great help.

Line 400. The authors put together traditional and underutilized crops. I only noticed this at this late stage. They are distinct and should be treated separately.

We also agree – we did this in the context of VACS as we indicated above – VACS clumps traditional/under-utilized/indigenous together. We need to clarify this. Perhaps we need a Box of definitions?

Oil palm, soybeans, sorghum, and canola are all traditional crops (there are many more but I will use these as examples.) If you go back to the period directly after World War II these were underutilized crops. So, at that time they were traditional/underutilized crops. Nowadays they are not underutilized (some would say oil palm is overutilized!) but surely in some areas they are traditional. Hence, to somehow make the leap to always putting the traditional and underutilized together doesn´t make sense (at least to me!). I wonder if rather than talking about traditional it wouldn´t be better to say of local origin or to be faithful to Vavilov crops within their centres of origin. The great Purseglove, who is mentioned above, commented on the disease escape of the industrial crops of the time and noted that many were perennials. This actually becomes very interesting. I think it was van de Planck who noted that horizontal resistance was more likely to develop in perennial species as they did not have a period to escape like annuals nor did they have a seed stage which, in come cases, eliminates diseases. (Potatoes tend to screw up this line of thought as I am not quite sure if they are perennials, or cultivated as annuals with vegetative propagation, but there are always exceptions.) It may be worth mentioning that, for these reasons, the introduction of perennials which are likely to have more durable genetic resistance is particularly attractive and the agroecologists should not deny that opportunity to the farmers.

Agree – we do need to at least mention perennials in this context. The importance of horizontal resistance in perennial crops was also of significant interest to Norman Simmonds who worked on breeding perennial crops. We had a memorable meeting in the UK in the early 1990s when I was working on disease management in mixtures of beans in East Africa and had on-going correspondence on this issue. We continued to disagree about the need for horizontal resistance in annual crops but it was an enjoyable interaction.

I return to the much maligned oil palm. It is amongst the most hated of the crops by the agroecologists and ambientalists. It is grown in monoculture, it is grown much more widely outside its centre of origin than within, it requires large amounts of inorganic fertilizer. However, from other points of view it is rather agroecological. Pesticide use is minimal. The canopy that it produces is much nearer to a tropical forest environment than let us say pastures or soybeans grown on cleared forest land or Alang Alang (Imperata cylindrica) grassy weed monocultures that often take over cleared degraded forest lands in SE Asia. While grown as a monoculture apparently, ground cover is provided often by leguminous cover crops which are not harvested as a crop, but do provide some biodiversity. Finally, and this point is not just relevant for oil palm. Using intensive “industrial” production methods oil palm produces much more oil per hectare than any other oil crop. Hence, if one considers the extra land required to grow other oil crops or that used to produce the same amount of oil with agroecological palm production, the area cultivated would have to expand enormously. This point is in the article and in the conclusions, but somehow it is hidden and does not get the credit it deserves. To this reviewer it is one of the most important drawbacks of the Agroecology approach and should be highlighted.

Agree – we need to emphasize this point and we like your arguments but I am reluctant to introduce oil palm into the paper due to the possibility that the criticisms of oil palm – not just by agroecologists but many agricultural scientists – will over-shadow some of the important points made in the paper.

Reviewer 2 Report

Comments and Suggestions for Authors

Overall, this paper is disappointing and needs significant reformulation to provide a much more balanced narrative, as explained below.

What the paper means by food “security” is not clearly defined, which is strange given that lack of this is one of the main arguments against alternative food systems. In fact, agroecology, as one of the latter, in addition to talking a lot about food security, also mentions issues like availability and accessibility which are central the food security for the consumer (e.g.; https://www.annualreviews.org/docserver/fulltext/resource/15/1/annurev-resource-102422-090105.pdf?expires=1737981046&id=id&accname=guest&checksum=590F353B5C5F543FAF62D9D46B23F755; and https://ec.europa.eu/programmes/erasmus-plus/project-result-content/ccc04415-ba38-431d-8b9a-8616f1c00949/O3Handbook_%20EN.pdf) According to the Global Strategic Framework for Food Security & Nutrition, the four pillars of food security are availability, access, utilization and stability. This paper does not mention these issues, neither individually not collectively, all of which are also well covered on the FAO’s Agroecology Knowledge Hub. This reviewer is not arguing for or against agroecology, but just pointing out inconsistencies in the paper and arguing for quite a lot of rewriting and reformulation.

In my specific comments, I have focused on issues that give me concern, of which there are so many that I have stopped commenting in detail after page 5, even though there are many good, valid and well-argued and/or well-referenced points across the paper.  

The paper seems to try to throw the agroecological baby out with the bath water, without clear analysis of how it can contribute useful ideas and practices in many contexts, alongside, and perhaps in place of, the “modern” types of agriculture that the paper seems fixated on promoting, whilst it attempts to demolish most things agroecological. In spite of this, the conclusion is good, emphasising that “the best conventional and agroecological approaches are well-suited to integration” but this is not evident in much of the paper.

A thoroughly rewritten balanced approach is necessary, that does not use emotion language and provides concrete arguments and/or good references for its statements. Indeed, your ref 78 concerning ecologically based monodominance, does attempt to do this when, for example, it concludes “There is still an opportunity to revise the current thinking to develop a more ecologically-based form of agroecology.” And there are strong movements in this direction, to give just one example in much of George Monbiot’s work -- someone who strongly supports agroecology, but is clear that it cannot feed the world on its own and should also move to what he terms “high-yield agroecology”, all based on extensive scientific research. This is also the main reason he advocates a lot of focus on – but by no means exclusively – technologies like vertical farming, lab-based protein production and GMOs when adequately regulated, to accompany extensive rewilding. Thus, one could say that agroecology has “no internationally recognized definition” but this is because it is morphing and innovating though without claiming any unique insight or solutions. Nuances like this, as well as many others, are generally lacking in the paper which could otherwise make a useful contribution towards a way forward. 

Some specific issues:

1)     Lines 46-50 these lines should be linked to one or more authoritative sources, especially when using words like “good”, “judicious” and environmentally responsible”. As these statements are the core of the paper’s argument about the status quo being better than agroecology, they need to be very carefully justified by good scientific arguments and sources. This is especially the case when using the word “successful” in line 52 in contrast to labelling agroecology as “unorthodox”. None of these words are scientifically useful.

2)     Line 52: Is it possible to obtain more up-to-date data on mixed crop-livestock systems than reference 18 that was published in 2010, 15 years ago, given that this is a critical part of the paper’s argument? In fact, this paper describes the farms responsible as small holdings and argues that these should be the first target for policies to intensify production, i.e. this is prescriptive which you condemn the IAASTD for in Box 1. This is a pejorative word to describe an argument for putting into practice a particular approach. 

3)     Lines 61-63: For the benefit of the paper’s argument, it would be very useful to give some examples of how the “activist organizations are communicating false information in regions where food systems urgently require improvement with existing modern technologies”. In this context, it would be useful to explain the use of word ‘activist’, e.g. is it commonly used to describe a certain type of individual or organisation pushing certain agendas? Does it imply that other types of individuals or organisations are ‘not-activist’? In line 399, activists are equated with UN Agencies which is difficult to comprehend without some explanation. This reviewer is not necessarily arguing for the deletion of the word ‘activist’ but it seems to be being used pejoratively and as a general label, whereas it might be more desirable from an objective scientific perspective to use one or more specific descriptors.

4)     Lines 79-88: I thoroughly concur with the sentiments here, especially the need for a non-binary and pragmatic approach which is science-based, etc. But, it is a shame this good advice is generally not adhered to in the paper, e.g. use of words like “based on beliefs and anecdotes” and “ill-informed” are unfortunate as they are generally not adequately justified. There are many other unfortunate words and phrases in the paper so that some of it comes across as at least partially a diatribe.

5)     Line 95: There are generally accepted and respected definitions of agroecology, such as that internationally-recognised by the FAO (with its ten elements). However, it is correct there are many other definitions, although these tend to be context-based and have most of these elements in common. The type of food systems this paper argues for is clearly just as diverse in its definitions, probably more so, so this is a weak argument at best.

6)     Line 108: Amongst many other reputable sources that provide strong scientific justification, monoculture agriculture’s disadvantages are described in a Science Direct review (https://www.sciencedirect.com/topics/agricultural-and-biological-sciences/monoculture) most of which are environmental, with most advantages being commercial. Clearly this is not a good balance. One solution is rotation, which both the paper and agroecologists argue for. The paper is correct, however, in mentioning anti-GM as having no scientific justification.

7)     Line 109: Describing “agroecology is a “curate’s egg” – good in parts” is probably a fair description but, if so, it is certainly equally fair for most other types of food production, including the type that the paper argues for, presumably as described in lines 46-50

8)     Lines 111-113: There are two different references for (4), i.e. IAASTED ?? and McIntyre et al. If they are the same this should be made clear and consistency should be applied as it is often difficult to follow the argument and links to references. See also point 12 below.

9)     Lines 134-139: As mentioned above, all these sources are 14-15 years old, which means they are quite out of date in such a fast changing field.

10)  Box 1:

·         It is not clear what are the precise numbered references for the two quotations. 

·         I don’t find phrases like “a bridge too far” and “an extreme agroecological approach” scientifically useful. There should be more objectively neutral, evidence-based ways to make such criticisms.

·         There is indeed a debate about the tensions between sustainability and intensification/extensification, which many sources are now arguing could partially be addressed by new systems like vertical and precision farming and even lab-based production – which is also happening in Africa -- so it would be useful to rehearse some of these arguments if you wish to cite extensification.

11)  Lines 148-149: The phrase a “serious problem of bias” is not demonstrated in the text following Table 1. The fact that LAC mentions “agroecology” and/or “agroecological” much more than the other regions could simply mean these terms have been adopted much more here. What precisely is the bias?

12)  Line 161: Following on from the unclear references in Box 1, the precise reference for the “IAASTD Global Report” is not given.

13)  Lines 174-178: I find this paragraph perplexing. The fact that there is “lack of interest” (except in LAC presumably) is not an argument against those who promote agroecological approaches to continue to do so, nor to describe this as “archaism”. It might be argued that this very lack of interest is a very sound and rational reason why those supporting such approaches should carry on. Lack of interest says nothing about the validity or not of the science for and against, as many examples in the history of science demonstrate.

14)  Lines 179-184: This paragraph misses the critical issues of availability, access, utilization and stability for food security as mentioned above, all of which are well documented on the FAO’s Agroecology Knowledge Hub, but not addressed at all in this paper. The paragraph, as in many places elsewhere, also uses pejorative terms like “questionable” without providing evidence.

15)  Pages 190-191: It would be useful to provide a clear reference for the statement that the central tenet of agroecology is the belief that diverse species mixtures are the only solution to insect herbivory of crop fields.

Author Response

Responses to Reviewer 2:

Overall, this paper is disappointing and needs significant reformulation to provide a much more balanced narrative, as explained below.

We are sorry that the reviewer found the paper disappointing especially as the other reviewer was very positive about the paper. We disagree that it needs significant reformulation.

In response to the invitation to submit a paper to a Special Issue on “Emerging trends in Alternative and Sustainable Crop Production” we explained to one of the Guest Editors that we planned to write a critical analysis of several alternative trends that were not likely to contribute to science-based sustainable crop production. This was accepted by them and the publisher.

As the paper clearly states we focused on a critical analysis of agroecology and the Vision for Adapted Crops and Soils (VACS) in the context of the Global South. We did not plan to look at other areas of the world where some forms of agroecology are more or less the same as Good Agricultural Practices e.g. Europe.

There is a comprehensive literature covering the agroecological approaches being promoted in the Global South, many of which are collated in the 600+ page report of the IAASTD (ref. 4). Some of these have been synthesized into the FAO Elements and HLPE Principles. This was the main focus of our critical analysis of agroecology. Its criticism by the World Bank Evaluation Group in 2010 (the World Bank partly funded the IAASTD process at USD 12 million) is sufficient reason to justify a critical analysis. The fact that VACS is a selection of some of the recommendations in the IAASTD reports also justifies its inclusion in this critical analysis.

We will try to make this point clearer in the paper – we are not criticising all agroecological approaches – we are criticising those being promoted in the Global South with emphasis on Africa.

In addition, both authors have published previous papers on agroecological approaches and we wanted to ensure as far as possible that we did not repeat what we have already published.   

We have also considered your comments on the use of “emotive” language which depends of subjective judgement – what emotes you is different to what emotes other readers. We have attempted to address some of the terms you disagree with but others – which are taken from the references cited – have stayed in the paper. We especially disagree with you in relation to the use of the term “belief” in the context we have used it. A “belief” is an opinion, hope, faith, assumption, conviction, view, position – I could also list other synonyms. The agroecology being promoted in the Global South is based on a belief that it will produce as much food or more food than existing crop production systems. As the evidence to support this is limited, it has not been proven and therefore it is a belief.

Finally, we make no excuse for using some older references which are key to the arguments we use in this paper. Too often such research is either forgotten or ignored by younger scientists leading to scarce research funding being used to repeat past research with the same results. Many of the references cited in this paper are from the past 4-5 years.

What the paper means by food “security” is not clearly defined, which is strange given that lack of this is one of the main arguments against alternative food systems. In fact, agroecology, as one of the latter, in addition to talking a lot about food security, also mentions issues like availability and accessibility which are central the food security for the consumer (e.g.; https://www.annualreviews.org/docserver/fulltext/resource/15/1/annurev-resource-102422-090105.pdf?expires=1737981046&id=id&accname=guest&checksum=590F353B5C5F543FAF62D9D46B23F755; and https://ec.europa.eu/programmes/erasmus-plus/project-result-content/ccc04415-ba38-431d-8b9a-8616f1c00949/O3Handbook_%20EN.pdf) According to the Global Strategic Framework for Food Security & Nutrition, the four pillars of food security are availability, access, utilization and stability. This paper does not mention these issues, neither individually not collectively, all of which are also well covered on the FAO’s Agroecology Knowledge Hub. This reviewer is not arguing for or against agroecology, but just pointing out inconsistencies in the paper and arguing for quite a lot of rewriting and reformulation.

We agree with the reviewer’s comments on the use of the term food security especially after reading Ken Giller’s 2022 paper “The Food Security Conundrum in sub-Saharan Africa”. Global Food Security 26: 100431. Food security is a wider concept than our focus in this paper. We have changed the title to: “The promotion of alternative sustainable crop production paradigms should be founded on science-based approaches”. We have also replaced the term “food security” with “food production” and other relevant terms throughout the paper.

In my specific comments, I have focused on issues that give me concern, of which there are so many that I have stopped commenting in detail after page 5, even though there are many good, valid and well-argued and/or well-referenced points across the paper.  

The paper seems to try to throw the agroecological baby out with the bath water, without clear analysis of how it can contribute useful ideas and practices in many contexts, alongside, and perhaps in place of, the “modern” types of agriculture that the paper seems fixated on promoting, whilst it attempts to demolish most things agroecological. In spite of this, the conclusion is good, emphasising that “the best conventional and agroecological approaches are well-suited to integration” but this is not evident in much of the paper.

We return to the objectives of the paper (see above). We are not attempting to criticise all agroecological approaches – especially not those which are science-based. Many of these e.g. crop rotation and inter-cropping were developed long before agroecology was conceived. We have made this clearer in the paper. 

A thoroughly rewritten balanced approach is necessary, that does not use emotion language and provides concrete arguments and/or good references for its statements. Indeed, your ref 78 concerning ecologically based monodominance, does attempt to do this when, for example, it concludes “There is still an opportunity to revise the current thinking to develop a more ecologically-based form of agroecology.” And there are strong movements in this direction, to give just one example in much of George Monbiot’s work -- someone who strongly supports agroecology, but is clear that it cannot feed the world on its own and should also move to what he terms “high-yield agroecology”, all based on extensive scientific research. This is also the main reason he advocates a lot of focus on – but by no means exclusively – technologies like vertical farming, lab-based protein production and GMOs when adequately regulated, to accompany extensive rewilding. Thus, one could say that agroecology has “no internationally recognized definition” but this is because it is morphing and innovating though without claiming any unique insight or solutions. Nuances like this, as well as many others, are generally lacking in the paper which could otherwise make a useful contribution towards a way forward. 

Apart from the call to rewrite the paper in a balanced way – this was not the objective of our paper as noted above – we agree with your comments. When the term “agroecology” was coined 90 years ago by German scientists it referred to agronomy and pest management. Although it was largely forgotten until the 1970’s/1980’s, when it re-emerged it was a hybrid between gardening and farming. In the interim period, it morphed into a mixture of science, practices and social movements with no internationally recognized definition – open to interpretation by both rationalists and radicals. Unfortunately, this has provided opportunities by activists to promote whatever they wish to interpret agroecology as. The focus of this paper is on the dangers of this approach for the Global South. As you note, in the hands of scientists, innovation in agroecological approaches to food production can provide unique insights or solutions. Unfortunately in the hands on non-scientists, especially those who consider current modern agricultural approaches to be unsuitable for smallholder farmers in the Global South, widespread adoption of the dominant IAASTD options would lead to reduced productivity gains and more environmental damage through extensification – as concluded by the World Bank evaluation.

As much as I agree with George Monbiot’s and other’s work in the UK, much of what is recommended is not relevant to the Global South where the main objective of smallholders is the produce enough food to feed their families with very limited resources. Rewilding, vertical farming and lab-based food is not relevant to such farmers.

Some specific issues:

1)     Lines 46-50 these lines should be linked to one or more authoritative sources, especially when using words like “good”, “judicious” and environmentally responsible”. As these statements are the core of the paper’s argument about the status quo being better than agroecology, they need to be very carefully justified by good scientific arguments and sources. This is especially the case when using the word “successful” in line 52 in contrast to labelling agroecology as “unorthodox”. None of these words are scientifically useful.

References added; good agricultural practices, often referred to as GAP, is a term which has been used for over 20 years including significant promotion by FAO from around 2014. It is well-explained by terms including “judicious” and “environmentally responsible”.

We have removed “unorthodox”.

2)     Line 52: Is it possible to obtain more up-to-date data on mixed crop-livestock systems than reference 18 that was published in 2010, 15 years ago, given that this is a critical part of the paper’s argument? In fact, this paper describes the farms responsible as small holdings and argues that these should be the first target for policies to intensify production, i.e. this is prescriptive which you condemn the IAASTD for in Box 1. This is a pejorative word to describe an argument for putting into practice a particular approach. 

The Herrero et al. (2010) reference to food production in crop-livestock systems is the foundation reference for this important fact – it is well-cited including today. Subsequent references including those below have shown the same results. I have added one to please the reviewer.

Herrero M, Havlík P, Valin H, Notenbaert AM, Rufino M, Thornton P K, Blummel M, Weiss F, Obersteiner M (2013). Global livestock systems: biomass use, production, feed efficiencies and greenhouse gas emissions. PNAS 110 (52), 20888-20893.

 Herrero M, Thornton PK, Power B, Bogard J, Remans R, Fritz S, Gerber J, Nelson GC, See L, Waha K, Watson RA, West P, Samberg L, van de Steeg J, Stephenson E, van Wijk M, Havlik P (2017). Farming and the geography of nutrient production for human consumption. The Lancet Planetary Health 1, 33-42.

Kerr RB, Hasegawa T, Lasco R, Bhatt I, Deryng D, Farrell A, Gurney-Smith H, Ju H, Lluch-Cota S, Meza F, Nelson G, Neufeldt H, Thornton P (2022).  Chapter 5: Food, Fibre and Other Ecosystem Products. In: Climate Change 2022: Impacts, Adaptation and Vulnerability

https://www.ipcc.ch/report/ar6/wg2/chapter/chapter-5/

These clearly show the fact that most crop- and livestock-derived nutrients are produced in places where the systems are mixed – and in lower-income countries, that proportion is very higher.

Too often, foundation papers are ignored/not known about which leads to repetition of previous work – often claimed as novel.

There is a clear difference between the Herrero et al paper and the IAASTD report. The Herrero paper did not state that it would not make recommendations at the end of the modelling exercise that generated the results presented. Most, if not all, scientific papers would do so in the conclusions. In contrast, the IAASTD process and report stated categorically that it would not be prescriptive especially about future policies for food production. If you read the report it is very clear. The Word Bank evaluation judged that the IAASTD report had been prescriptive. All we are doing is stating what the Word Bank said – as Box 1 is attributed to this reference. It is their term – the World Bank is expressing disapproval of the IAASTD report.

3)     Lines 61-63: For the benefit of the paper’s argument, it would be very useful to give some examples of how the “activist organizations are communicating false information in regions where food systems urgently require improvement with existing modern technologies”. In this context, it would be useful to explain the use of word ‘activist’, e.g. is it commonly used to describe a certain type of individual or organisation pushing certain agendas? Does it imply that other types of individuals or organisations are ‘not-activist’? In line 399, activists are equated with UN Agencies which is difficult to comprehend without some explanation. This reviewer is not necessarily arguing for the deletion of the word ‘activist’ but it seems to be being used pejoratively and as a general label, whereas it might be more desirable from an objective scientific perspective to use one or more specific descriptors.

We have added more information about who some the activists are although there are many which are mentioned in previous papers by the authors. We have made sure that there is a distinction between FAO and the activists. As the majority of people know that an activist organization is one pushing a particular agenda then it would insult the readers’ intelligence to explain this. This paper is not a forum to debate what is or is not an activist organization. By implication, are you saying the genuine scientific organizations such as CGIAR are pushing an agenda rather than developing needed science-based soultions to increased food production.

4)     Lines 79-88: I thoroughly concur with the sentiments here, especially the need for a non-binary and pragmatic approach which is science-based, etc. But, it is a shame this good advice is generally not adhered to in the paper, e.g. use of words like “based on beliefs and anecdotes” and “ill-informed” are unfortunate as they are generally not adequately justified. There are many other unfortunate words and phrases in the paper so that some of it comes across as at least partially a diatribe.

Again, the terms used are not only used by us but by others in a number of papers and reports. We have attempted to justify use of the terms or modified them. For example: one of the key proponents of the diversity element/principle of agroecology uses the term “mantra” in support of diversity and he is a scientist.

However to promote alternative food production approaches with no proven ability to produce the same amount or more food than the existing proven approaches is by definition a belief and ill-informed and in some cases the justifications provided are anecdotal.

5)     Line 95: There are generally accepted and respected definitions of agroecology, such as that internationally-recognised by the FAO (with its ten elements). However, it is correct there are many other definitions, although these tend to be context-based and have most of these elements in common. The type of food systems this paper argues for is clearly just as diverse in its definitions, probably more so, so this is a weak argument at best.

We question whether the FAO definition of agroecology is generally accepted and respected – certainly not in my experience. If this was the case then why are there so many other definitions? On one website alone we found over 30 different definitions. Also, the paper is focused on challenging two of the FAO elements – diversity and input reduction – hence by association it is critical of the FAO definition and the elements.

The paper is strongly based on science-based solutions to future food production through improving existing systems – monoculture-based agriculture – which currently produces most of our food. At no stage do we argue for diverse systems in the same context of agroecology. We support functional diversity – both within crops and within fields.

6)     Line 108: Amongst many other reputable sources that provide strong scientific justification, monoculture agriculture’s disadvantages are described in a Science Direct review (https://www.sciencedirect.com/topics/agricultural-and-biological-sciences/monoculture) most of which are environmental, with most advantages being commercial. Clearly this is not a good balance. One solution is rotation, which both the paper and agroecologists argue for. The paper is correct, however, in mentioning anti-GM as having no scientific justification.

We have clearly stated in the introduction to the paper that: “It is true that some modern agricultural practices, especially in the Global North, including overuse of agrochemicals and mono-cropping, have led to environmental problems such as pollution, soil erosion and loss of wild biodiversity and need to be addressed with science-based approaches”. But these problems are well-recognized and “Substantial research effort is already underway to do so (13, 14, 15)”.

We have added in a following part of the paper that these include good agricultural practices which include some practices which have been renamed and subsumed under agroecology e.g. crop rotation, inter-cropping, cover crops etc.

7)     Line 109: Describing “agroecology is a “curate’s egg” – good in parts” is probably a fair description but, if so, it is certainly equally fair for most other types of food production, including the type that the paper argues for, presumably as described in lines 46-50.

We agree and we have raised this above and included the Balmford et al. (2018) (ref. 72) reference raises the same issue.

8)     Lines 111-113: There are two different references for (4), i.e. IAASTED ?? and McIntyre et al. If they are the same this should be made clear and consistency should be applied as it is often difficult to follow the argument and links to references. See also point 12 below.

There are seven IAASTD reports: global full report, global synthesis report, five regional reports – LAC, SSA, CWANA, NAE, ESAP. The first two are attributed to McIntyre et al 2009 (ref. 4); the regional reports are attributed to IAASTD (2009). You can check the citations on the website. We have made sure that all citations of the global report are accompanied by (4)

9)     Lines 134-139: As mentioned above, all these sources are 14-15 years old, which means they are quite out of date in such a fast changing field.

The references cited in lines 134-139 are critical to the analyses provided in the following section of the paper. On one hand you say they are out-dated and then you challenge them. Unfortunately the IAASTD (ref. 4) has NOT moved in. Ten years later it is still promoting the same agroecological approaches – see Herren et al. (ref. 24) also Frison (ref. 33).

It was clearly stated in the paper that we looked at the evolution of the concept of agroecology from when it was first coined in the 1930’s to how it has evolved over the past 90+ years. If you can suggest a way to do this without citing old references that capture these changes then please do so.

10)  Box 1:

  • It is not clear what are the precise numbered references for the two quotations. 

The heading of Box 1 cites the World Bank Evaluation report ref 30. It is clear where the quotations came from. Please check the report.

  • I don’t find phrases like “a bridge too far” and “an extreme agroecological approach” scientifically useful. There should be more objectively neutral, evidence-based ways to make such criticisms.

You need to take this up with the World Bank as this is exactly what they said.

  • There is indeed a debate about the tensions between sustainability and intensification/extensification, which many sources are now arguing could partially be addressed by new systems like vertical and precision farming and even lab-based production – which is also happening in Africa -- so it would be useful to rehearse some of these arguments if you wish to cite extensification.

Box 1 is clearly attributed to the World Bank Evaluation report of the IAASTD – ref. 30. Every word including the quotations is from their report. I do not think it is necessary to add (30) to the quotations when it is already in the title of Box 1.

We are reporting exactly what the World Bank concluded. Please check the report – this is their wording – we are just citing it. Vertical and precision farming are outside the scope of this paper as it is a discussion of what the World Bank Evaluation report concluded. I would also challenge the potential for vertical and precision farming as priority technologies for food production for many countries in the Global South – the main focus of this paper as stated in the introduction.

To open up a debate on extensification – cultivating more land for food production – is somewhat marginal to the main objective of this paper.

11)  Lines 148-149: The phrase a “serious problem of bias” is not demonstrated in the text following Table 1. The fact that LAC mentions “agroecology” and/or “agroecological” much more than the other regions could simply mean these terms have been adopted much more here. What precisely is the bias?

The bias refers to the Global report and is clearly demonstrated. We are not questioning the numbers of mentions of agroecology in the LAC report. What we are questioning is the strong emphasis on agroecology in the Global report which should represent the reports from all regions. It does not represent the findings of the SSA and CWANA reports. This is clear bias toward the LAC report in the Global report.

12)  Line 161: Following on from the unclear references in Box 1, the precise reference for the “IAASTD Global Report” is not given.

See above – explanation about the different IAASTD reports. The precise reference is McIntyre et al (2009) ref. 4. We have now made sure that all references in the paper to the Global report are attributed to McIntyre et al 2009 or ref. 4.

13)  Lines 174-178: I find this paragraph perplexing. The fact that there is “lack of interest” (except in LAC presumably) is not an argument against those who promote agroecological approaches to continue to do so, nor to describe this as “archaism”. It might be argued that this very lack of interest is a very sound and rational reason why those supporting such approaches should carry on. Lack of interest says nothing about the validity or not of the science for and against, as many examples in the history of science demonstrate.

What you are saying is that we should not listen to those in other regions who do not support agroecology but we should carry on promoting it even if they do not want it – e.g. SSA and CWANA and 67% of African countries. That is a top-down approach which is not acceptable in most countries today.

14)  Lines 179-184: This paragraph misses the critical issues of availability, access, utilization and stability for food security as mentioned above, all of which are well documented on the FAO’s Agroecology Knowledge Hub, but not addressed at all in this paper. The paragraph, as in many places elsewhere, also uses pejorative terms like “questionable” without providing evidence.

We addressed this above. We agree – food security is a complex issue which cannot be addressed by food production alone. We have changed the title to reflect this and reworded other parts of the paper.

15)  Pages 190-191: It would be useful to provide a clear reference for the statement that the central tenet of agroecology is the belief that diverse species mixtures are the only solution to insect herbivory of crop fields.

Fair enough – we have modified the wording.

Altieri has published hundreds of papers on the topic basically saying that diversity is the best way to control pests in agroecosystems. We included:

  1. Altieri, M. A. Nicholls, C. I. Montalba, R. Technological approaches to sustainable agriculture at a crossroads: An agroecological perspective. Sustain. 2017. 93, 349. http://doi.org/10.3390/su9030349 later in this section.

How many more of his papers are needed when a number of meta-analyses have shown that this is not correct – the results are inconsistent to say the least.

Round 2

Reviewer 2 Report

Comments and Suggestions for Authors

See attached comments in Word document

Round 3

Reviewer 2 Report

Comments and Suggestions for Authors

As said before, I am now happy that the paper should be published – I am assuming the other reviewer thinks so as well – especially now that further improvements have been made.
